# Responding to Autism in Low and Middle Income Countries (Lmic): What to Do and What Not to Do

**DOI:** 10.3390/brainsci12111475

**Published:** 2022-10-30

**Authors:** Roy McConkey

**Affiliations:** Institute of Nursing and Health Research, Ulster University, Belfast BT15 1ED, Northern Ireland, UK; r.mcconkey@ulster.ac.uk

**Keywords:** autism, community-based, community support workers, family-centered, low and middle income countries, support services, training of trainers

## Abstract

Autism is a spectrum disorder that occurs globally with increasing numbers of children and adults being identified with this condition. Although rates are higher in more affluent nations, the bulk of people with autism reside in Low and Middle Income Countries (LMICs). However, most do not have access to timely and appropriate support. The types of services delivered in high income countries are often ill-suited to the needs and resources of LMICs. Rather alternative forms of provision need to be devised. In common with other public health initiatives, these should be family-focused and community based, with suitably qualified and experienced leaders who in turn train and supervise a cadre of knowledgeable support workers drawn from local communities and cultures. As well as providing personalized, home-based guidance to people with autism and to their primary carers, regular group-based advocacy and training activities are undertaken in partnership with available mainstream services such as schools and social services. The principles and operations of these new forms of services are described in this paper albeit with an acknowledgment of their limitations. In recent decades, the cost-effectiveness of these approaches have been demonstrated with other chronic illnesses and disabling conditions in LMICs but their extension to autism has barely begun. More affluent countries are being forced to adopt similar strategies in response to the increased numbers of people identified with autism. A transformation in research strategies is essential to building better international support for persons with autism.

## 1. A World Apart

Autism affects children and adults in every nation and probably has done so since humans socialised and communicated. However, in the scope of human history, autism is a modern condition; a product of scientific advancement and affluence [1]. The term autism came to the fore in the 1940s with an Austrian psychiatrist—by name Leo Kanner—who noticed a similar constellation of symptoms that were present in some of the young children referred to him. Around the same time, another Austrian doctor— Hans Asperger—came to a similar conclusion although these were children with less severe effects. It was not until his writings were published in English in the 1980, that the term Asperger’s Syndrome began to be used for those with milder forms of autism.

The post war period especially from the 1960s onwards with its increasing affluence, heralded an expansion of university education throughout Europe and the United States in particular. The growth of specialities in medicine such as paediatrics and psychiatry along with new disciplines, for example psychology and the therapies, resulted in increased research into children’s development and innovations in professional practice. The development of new tools for assessing children’s abilities led to more differentiation among children who might previously have been described under catch-all terms such as ‘slow learner’, ‘mentally retarded’ or ‘mentally ill’.

Autism started to become a distinct condition in the 1980s when the American Psychiatric Association defined its symptoms in their Diagnostic and Statistical Manual of Mental Disorders” (DSM, Third edition) and internationally from 1990 onwards by the World Health Organisation in their International Classification of Diseases, 10th Edition. In recent years, both these classifications systems have updated the criteria for classing a child as having autism (DSM-5 [2]; ICD-11: [3]). The main change is that both systems promote the concept of a range of autism spectrum disorders rather than a single condition. To further complicate matters, autism can occur alongside other conditions—referred to as co-morbidities—such as intellectual disability or a mental illness such as anxiety. It is widely recognised that an accurate diagnosis of autism is most likely to be achieved through multi-disciplinary assessments using ‘gold standard’ assessment tools [4]. Nonetheless, there are wide disparities across nations in the prevalence of children with a diagnosis of autism with the highest rates occurring in some of the most affluent countries and the lower rates usually reported in Low and Middle Income Countries (LMICs) [5].

Diagnosis needs to be followed by treatment and over the past four decades many novel interventions specific to autism have evolved although the merits of each approach have been hotly debated [6]. However, what they mostly have in common, is a focus on dedicated personnel trained in autism delivering the ‘treatments’ over a period of time, often at additional expense to the family or governments through their health and education systems.

From a global perspective, these developments took place solely in affluent, English-speaking countries and were a world apart from the daily reality facing families and health services in LMICs as they struggle to support children who are experiencing developmental difficulties, including those resulting from autism. LMICs typically lack experienced personnel across all disciplines, a scarcity of training opportunities for existing and new staff and the poverty of resources to fund new, let alone improve existing services. More profoundly, we have a greater appreciation of the cultural variations and societal contexts surrounding children and families, and the impact these have on their understanding of, and responses to developmental difficulties [7]. Hence, it is unrealistic and possibly counter-productive to insist that LMIC health and educational services should emulate the procedures and practices developed in and for affluent countries.

However, that is not to imply we should ignore the new knowledge and insights about autism that have been gained through the efforts of more affluent nations. They have generously shared them through written publications which have become ever more accessible world-wide thanks to the Internet and mobile phone technology. Rather we must learn how to apply the available knowledge about autism into low-resourced settings and with differing cultures so that all the world’s children with autism—the vast majority of whom live in LMICs—can receive at least a modicum of support to live more fulfilled lives. A right—we should remind all governments—that is made clear in the United Nations Rights of the Child and of Persons with Disabilities to which most nations are signatories.

Equally, LMICs can bring fresh insights into our understanding of autism; notably the broader social, economic and cultural influences that impact on the condition and the low-cost responses that can ameliorate its impact on the lives of children and families. Sadly many researchers and professionals in affluent countries remain to be persuaded of the value of understanding the social and cultural influences on developmental disabilities and reflecting this in their theorizing and practice [7]. This is all the more surprising given the inward migration of people from other countries and cultures into affluent European countries, the USA and Canada. The new arrivals in time become citizens and yet those whose children have autism or other developmental disabilities, face many difficulties in accessing services that take account of their linguistic and cultural differences [8].

This paper presents a personal thematic perspective on lessons learnt and pathways for future actions in LMICs. The themes identified in the review come from nearly 40 years experience of my working alongside colleagues, some of whom are named in the acknowledgements but others are too numerous to mention. The insights are supported by the experiences documented in the provision of community-based health, education and social services in LMICs that are referenced below but to my knowledge this is the first attempt to apply them to autism with focus particularly on creating low-cost, support services for children with autism in particular and for their family carers.

## 2. What Not to Do

The good news is that we are discovering effective ways for LMICs to support families and children with autism. As these become more widespread even more approaches will be invented. The less good news and in some ways even more challenging, is that we have to change national beliefs about autism in particular and stop doing certain practices that at best are likely to be ineffective and may even be harmful when misused.

Ironically, a similar issue is now arising in more affluent countries too, as the numbers of children and adults considered to have autism has grown rapidly through the sophisticated and costly diagnostic and assessment processes professionals use. Yet, the financial resources to provide ongoing supports after diagnosis are constrained and insufficient to meet the increased numbers of people needing them. Given the financial resources available in rich countries, their choice might be better framed as what they should spend less money on and where more money should be spent [9].

Here, are my five top tips for what LMICs should NOT do as a response to autism.

### 2.1. More Professionals Are Not the Solution

Creating more highly trained professionals to meet the needs of people with autism is a favoured response but it is not suited to LMICs in particular [10]. It requires an investment in specialised training courses across different disciplines that are costly to set-up and which take four years or more to complete before a return is had on this investment. Even then, there is no guarantee that graduates will take up employment in their country and some will emigrate to find better opportunities. Those who stay will expect high salaries and many will enter private practice which only richer families can afford. They prefer to be based in major cities which leaves people in rural areas bereft of support. Admittedly, some professionals with expertise in autism are needed as leaders and educators as will be discussed later but this should be done in the context of creating a broader workforce that can respond to the ongoing needs of children and families on a regular and local basis [11].

This may require a major shift in the expectations of those parents who may equate autism to an illness that requires special treatments from professionally qualified persons. Many find it hard to accept that autism is a life-long condition and children will require ongoing support that is best available to them on a daily basis [12]. Weekly sessions with some professionals and even daily ones with teachers are no substitute for what families can achieve when they are trained and guided on supports that will help their child’s development [13].

### 2.2. Specialist Therapies for Autism Are Not Always Needed

After nearly 40 years of debate, it is clear that no one therapy is effective for all of the children with autism, all of the time. One particular controversy has been around Applied Behavioural Analysis (ABA) with its proponents insisting it is the most effective means of assisting children with autism. Undoubtedly it has been effective with some children in some settings but it is also resource intensive and issues of its costs have precluded its adoption in many services systems [14]. Equally other interventions have been developed with growing evidence for their effectiveness [6]. A point has been reached when it is possible to match certain interventions to the children’s particular needs. Moreover, persons who are familiar with the child—such as parents and teachers—can be taught the techniques so that they can be used in the natural environments of the home and classroom [15].

### 2.3. Do Not Set Up Specialist Autism Provision

Following on from the previous two points, there is the temptation to create a distinct service for children with autism to offer assessment and interventions. In affluent countries, families and professionals have successfully lobbied for such specialist provision but they come at a high cost. Nor is it solely in terms of money but also in depriving children with other developmental difficulties of receiving help, of creating waiting lists once demand exceeds supply (which invariably happens) and focusing delivery on certain locations, often main cities leaving rural communities without support [16]. Perhaps the biggest risk, is that existing health and education services no longer feel they have a responsibility to support children with autism when they believe they should be referred for specialised assistance. Admittedly, mainstream services may feel ill-equipped for this task but therein lies the solution. The focus has to be on building their confidence and expertise. I will return to this point in a later section.

### 2.4. Do Not Focus Only on the Child with Autism

Autism is a condition that children may have but the disabilities they experience arise also from the contexts in which they live, notably home and local communities, such as schools. For example, most will experience problems with communicating and socialising with their parents, extended family and other children. Focusing only on the child in a clinic setting or through one-to-one teaching is less likely to help them to become better communicators with other people or more able to socialise with them in other places. A better approach is to centre the support in real life settings that engage other people who interact with the child. In a sense, the people with whom the children interact become recipients of the intervention and through them, support is provided to the child to help overcome the disabling effects of autism [17]. This model of family-based or classroom based support is a promising alternative to the child-focussed interventions that have dominated responses to autism in more affluent countries.

### 2.5. Do Not Perpetuate Wrong Ideas about Autism

There are three common myths about autism but still widely held in many countries [18]. “*All children with autism are the same so they need the same support*”. This is patently false as even two children in the same family with autism can have very different needs. Moreover, the recognition now that autism takes the form of a spectrum of conditions reinforces the individuality of each child’s needs.

“*We need to make a diagnosis before we start treatments*.” The children’s developmental difficulties existed before the diagnosis and nothing changes because a diagnosis has been given. Providing help to children once concerns have been raised around communication, socialisation and/or behaviour management, is unlikely to do much harm but can do much good for worried parents [19].

“*It is waste of money to help these children*”. How is it a waste of money if the children’s and family’s problems are reduced and most often they can be? Indeed this claim becomes a self-fulfilling prophecy if help is not provided on the grounds that it would be wasted. However, *not* helping the children and families risks incurring extra costs if the person or family members remain unable to earn a livelihood or becomes a risk to others through antisocial behaviours [20]. Nonetheless, the implicit message of this myth could be that other children are more deserving of help. This is when arguments about rights and equal opportunities come in.

## 3. What to Do

By now you will have gleaned some of the approaches that may be particularly relevant to people with autism in LMICs and which are described in more detail in this section. Essentially, what is envisaged is the development of a support service to persons with developmental difficulties and to their families, with dedicated staff, its own budget and with defined policy and procedures. This service may be linked to, but is separate from, the generic public health services available to all children and families and from educational systems. It could be funded by government but delivered by a non-governmental agency (NGO) or the service could be hosted within a governmental system such as health, community development, or social services. These services may support both children and adults but for simplicity, I will refer to children. The same ethos and procedures can be applied to supporting adult persons albeit that their needs may be different; for instance, in supporting them to find gainful employment.

I should emphasise that the style of support service that is proposed, is not necessarily a cheap option but rather the rationale is that these approaches make better use of available finances. There are a growing number of examples of these new services being successfully established for children and adults with mental health problems [21] and various developmental conditions [22]. Indeed, these strategies are being advocated also in more affluent countries in order to provide more equitable and sustainable support to all those in need and as a means of overcoming the deficits in current provision [23].

In many LMICs some services modelled on those provided in affluent countries have been established. My plea to them is to undertake a radical appraisal as to how their services might be transformed to embody the ethos and strategies outlined in the following sections. I admit though that this can a harder road to travel not least as it involves changes in parental expectations and the roles played by staff. However, transformations can be done through visionary leadership, clear communication among the stakeholders and detailed planning [24]. In some ways though, it is easier to create new support services in locations where none are currently available.

### 3.1. Guiding Principles

The starting point is a re-appraisal of the principles on which national responses to autism are based. This is sometimes referred as the values or beliefs that guide the decision-making process within LMICs. Of course they will also need to harmonize with each nation’s cultural and political systems. They may include some of the following principles that currently feature in designing health and educational systems that are relevant to LMICs [25,26]. I should stress that principles are aspirations; they set a direction for travel even though much effort has to go into achieving the desired outcomes.

### 3.2. Giving Some Help to Many Persons in Need Is Preferable to Giving Only Some Persons a Lot of Help

This philosophy is founded not just on the basis of equity that everyone who needs help can access it but there another pertinent rationale. Prompt intervention can prevent children’s and family’s difficulties becoming worse and as we do not always know who are at greatest risk, it is better to make some support readily available to everyone with an expressed concern or need. Additional supports can be added on a phased basis with those who have greater needs, getting more support. It has been described as ‘tiers of support’ with many people at the base and decreasing numbers on the higher tiers and only small numbers receiving the most intensive (and expensive) supports [27].

### 3.3. All Children with Developmental Difficulties

What might be termed basic supports should be available to all children experiencing developmental difficulties and not be specific to those with suspected autism. Unlike more affluent countries, LMICs do not have the resources to create specialist teams for differing impairments nor is this a feasible option away from major cities. However, the argument is not just on the basis of cost. Children with differing developmental problems have many issues in common, mostly notably in communication and social behaviours. By focusing on the children’s functioning rather than their impairments, broadly the same interventions can be used with many children although with some adaptation to take account of any physical, sensorial or behavioral difficulties that they may have [28]. For those children who require more specialized support, specific interventions can be devised such as assistive devices to cope with visual impairments.

### 3.4. Family-Centred

A key principle is that services are provided to support families in their child’s care and development. In many LMICs, the family have been, and will continue to be, the primary carers of their children and often their care-giving extends into their adulthood. This principle brings a duality of purpose in that the family are also the focus for support that addresses their emotional, practical and informational needs as well as interventions and supports to help the child [29]. Note the focus is on families and not just parents so that siblings, grandparents, uncles and aunties can be engaged too. Family ties are arguably stronger in some societies and cultures that should make this principle more attainable. Nonetheless, the stigma surrounding disability may need to be challenged within some families which might form the initial support that is offered to them [30]. It is preferable that much of the support is provided in the family home or close by to it; a reminder to everyone that the children’s inclusion in the family is a driver of developmental progress.

### 3.5. Community-Located

In every country, nearly all families live alongside others in a community with shared resources such as shops, places of worship, health clinics, schools and so forth. Children with autism and their families—mothers especially—need support to interact with their peers and others in those communities. The gains can be plentiful in terms of the children’s development and providing parents with practical as well as emotional supports. Hence, the support systems must be developed within local communities and if possible using existing resources as opposed to locating them in more remote settings such as hospitals. One outworking of this principle is that the main supporters will then be drawn from local communities who then receive training and ongoing monitoring of their work. This theme is further developed below.

### 3.6. Partnerships

A single support service cannot be expected to meet the diversity of needs among children with developmental difficulties and those of families, in even small communities. Instead, the goal is to link the children and families into other potential supports that could be mobilized within communities. Hence, the core support service for children with autism (and other developmental disabilities) and their families, needs to forge partnerships with other agencies such as playgroups, crèches, schools, health clinics and so on [31]. This may involve the provision of training to staff working in such settings to increase their knowledge and skills in coping with children who are different [32]. Admittedly it is not easy to persuade people to take on what they may perceive as extra work which is further compounded by the stigma associated with disability. However, such community partnerships have been shown to strengthen communities to the benefit of all its members. The investment of time and resources is worthwhile.

## 4. Operational Issues

The new style of support service will also be distinctive in the way it operates although it will have much in common with other forms of community-based, public health initiatives or community-based rehabilitation projects [33]. Its operations may need to be tailored to available resources and the availability of personnel in the chosen locality in which it is based. In this section, some of the key operational procedures are identified. However, the form of the services, its policies and practices are best done through a consultative process with the stake-holders including persons with autism; albeit that their advocacy and empowerment is rarely encouraged [34]. Even then, the process has to start with a person or group of people who have the responsibility or wish to develop a support service. This could be a government official, an NGO or some concerned parents.

### 4.1. Leadership Group

The planning for the service in a designated locality should be undertaken by persons who will use the service, interested professionals such as doctors or nurses who are involved with children and families, and potential partners in its work, such as social services. Community leaders such as local politicians, chiefs and religious leaders should also be recruited. They could form the leadership team that prepares the business case for the support service and seeks funding for it. In many countries, the initial funding comes from charitable trusts or from individual donors. A commitment from government to fund some or all of the costs should be sought. The scope of the service may need to be attuned to available resources and address particular needs, such as a focus on early childhood from birth to around eight years [25]. As the service gets underway, this group could transform into a Management Board that oversees the running of the service.

### 4.2. Leaders with Expertise in Autism

The new service needs to be led by someone with experience and expertise in developmental difficulties, particularly autism and/or family support services. Their role is not just to administer or manage but more importantly they will be committed to the key principles of the service and able to build partnerships with families and communities. Crucially they will develop the knowledge and skills of the support staff through example and engagement with service users as they continue to supervise and monitor the work of staff. This is often termed ‘practice leadership’ [35].

Individuals with these leadership qualities may be hard to find locally but several options can be considered. People might be recruited from another locality to set up the new service, perhaps on a short-term basis. A person from the local area who seems promising can be sponsored to undertake additional training or placements with similar services in other areas or neighbouring countries. NGOs and disability organisations could also assist in identifying suitable candidates including those from other countries.

### 4.3. Recruiting, Selecting and Training Community Support Workers

The service will be delivered through community support workers (although other job titles might be more suitable). In the main, they will be people familiar with the local community and who have had personal experience of disability within their families. Indeed parents of children with disabilities have successfully fulfilled these roles with the bonus of providing an income for their own family. Potential recruits may also have been employed in other community services. In sum, these staff will be recruited more for their personal qualities and experience rather than on academic attainments although basic literacy skills in reading and writing are essential. The personnel specification will reflect these qualities.

There can be advantages to having staff employed on a part-time basis as this enables a wider range of workers to be employed and might reduce travel costs and time if they come from different parts of the locality. Invariably the stipends for these workers is less than those paid to qualified professionals such as teachers and therapists. Hence, available funding can mean that more staff can be recruited to the service.

However, they will need on-the-job training to prepare for the range of tasks with they will be expected to fulfil according to the job description. The service leader will design and deliver the training which could take various forms including one-day workshops, e-learning modules and peer-mentoring focussing on different topics and aspects of their role. Throughout the emphasis should be on practical activities that prepare staff for the activities they will undertake with children and families. Ongoing supervision and monitoring of the staff’s work can be achieved through regular one-to-one sessions as well as staff meetings. Consequently, these staff have been termed a ‘mid-level’ worker, between those holding a professional qualification and a basic health worker or assistant [36].

### 4.4. Provision of Resources

Modern technology makes it easier for staff to maintain contact with families (and vice versa) through the provision of mobile phones. Additionally, with an internet connection, staff can have access to written guidance, video programmes and pictorial resources for use in their work or for sharing with families. Record keeping can also be facilitated through computer-based systems which can be accessed by staff either at a central location or remotely through an internet connection. Although these services may not be widely available at present across LMIC countries, they are likely to become so in the near future [37]. Staff transport costs such as taxi fares may also need to be covered to facilitate home visits. Additionally, community projects based in small localities provide motor or pedal cycles for the staff to use. Meetings with groups of parents, for example training courses for staff can be held in community venues, such as schools, places of worship or traditional meeting venues. However, a case can be made for establishing a base—sometimes called a resource center—at which training courses can be held, act as a drop-in centre for parents to meet one another and where groups of children can take part in learning activities [38].

### 4.5. Advocacy and Empowerment

A key element of the service is to foster the empowerment and advocacy of families and people with disabilities. A well-used means for doing this is through the formation of a parents and friends association [39]. Meetings are held on a regular basis to build group solidarity and contain a mix of information giving as well as social activities. Community support workers may facilitate these meetings initially, but the goal would be for members to take on the leadership and organisational roles. The numbers attending initially may be small, but parents often gain a lot of emotional support from meeting other parents. These associations have also been to the fore in tackling the stigma around disability and arguing for equal opportunities for their children.

### 4.6. Demonstrating Success

The service needs to become visible to prospective users and community partners. Usually this takes the form of a building but when services are based in the community, other means of the publicising its work need to be used. This includes preparing attractive publicity leaflets and posters about the service and its contact details. Staff could give talks about their work to various community groups: such as school staff meetings, religious gatherings, community councils, sports clubs and so on. The staff might invite a parent to join in the presentation or older persons with a disability. Open meetings might be organised to debate a particular issue with invited speakers. The newspaper and radio stations can be invited to publicise the meeting in advance and report on it afterwards. The producers of radio programmes might be approached about having a magazine programme for parents with a phone-in option. The underlying messages are one of hope: people with a disability can learn new skills and support is available for families [40].

### 4.7. Keep Records

Services need to be able to evidence the impact of their support on children and families [41]. Keeping records of the children’s needs, their level of functioning and measuring changes over time are essential alongside details of supports provided, although this is easier said than done. The records are often brought together in the form of a Personalised Plan. This information allows the staff to adjust the support they provide but it also provides evidence of the effectiveness of the service. Similar information can also be gathered about the impact on families—especially mothers if they are the main care-giver. Computerised recording systems are available for use on tablets or smart phones although these may not be immediately available in LMICs. However, the bigger problem is ensuring staff maintain the necessary records and that systems are place to enable reports to be drawn down for the information collected.

## 5. What Community-Based Services Struggle to Do

Although the approach proposed for responding to the needs of people with autism in LMICs has much merit, it is only fair to acknowledge its limitations [42]. These by no means invalidate the need for community-based approaches but rather they highlight the additional challenges that remain to be faced in the coming years and decades.

### 5.1. Widespread and Uniform Coverage across the Country

Because the support service is locality based there is inevitable variation across the country. In some communities the service may be quickly established due to the enthusiasm and expertise of local people on the planning group and the staff they recruit. By contrast other localities, often in poorer and more rural areas, lack these resources so it can be difficult to establish such services or attempts to do so may wither away. The consequence is that a patchwork of support for children and families emerges. This is a common phenomenon in health and social services, even in affluent countries but that is no excuse for perpetuating these inequities [43].

One approach would be to develop a national network of local services that is affiliated with an existing national governmental structure, for example a Ministry for Community Development or Social Affairs. The community-based service network would advise and assist the Ministry to seed services in other localities through their sharing of their expertise in service planning and delivery, staff training and parental empowerment. National policies and laws relating to service provision for persons with disabilities would facilitate such developments. Progress may be slow and perhaps national coverage will never be attained but it should not deter attempts to try.

### 5.2. Meeting the Needs of Persons with More Severe and Complex Needs

A small proportion of persons with autism and other developmental conditions would benefit from more specialised advice and treatments. Just as in richer countries, their needs often go beyond the competence of local staff. Ideally there should be opportunities for community services to refer persons to specialists such as therapists and doctors but such persons are scarce and over-worked in many LMICs.

Another function of the coordinating Ministry proposed above, would be the development of a national panel with a multi-disciplinary membership which local services could consult about specific individuals and receive training on how they might handle them. With the advent of better tele-communications, such consultations could take place via phone or video conferences so that travel costs are avoided [44]. More generally, training courses on common topics could also be provided by the panel again face-to-face or by video conferences. These could be open to family members as well as community staff.

Note that these approaches focus on knowledge, skill and resource transfer from the specialist to the people who have regular contact with the child and family. The panel members would NOT take referrals from community services—although occasional exceptions could be made—but it should never become common practice. Responsibility for helping these children and families must be retained by the community service and not passed to others. In due time, the competence of local personnel increases as the new expertise they have gained can be applied to other individuals.

### 5.3. Turn-Over of Staff

One of the frustrations for managers of community-based services is staff leaving for other jobs. This is especially irksome given the time and effort that has gone into their training which has to start again with new recruits. Likewise, the relationships they have built with children and families are not easily replaced. A two-strand approach could be adopted. First, aim to recruit persons who are more likely to stay with the service. This could include people from that locality who have commitments that will keep them there, such as parents raising children, active retired persons or persons with disabilities. All of these examples I have given can be found in community-based services internationally.

Second, consider the incentives that staff can be given to stay working in the service. Their stipend or salary could be increased the longer they continue working—in the form of an annual bonus for example. However, often job satisfaction is the key to better retention. Managers need to ensure that staff are recognised and appreciated for their work and there are myriad ways for doing this but giving praise rather than rebukes is probably among the most effective [45].

### 5.4. Maintaining and Increasing Funding for Services

Finding sufficient money to fund community services for persons with disabilities is a world-wide problem but one that is acerbated in LMICs because of the many other demands from other health, education, social and community services [46]. Money is less of a problem for richer families who can afford the fees charged by private practitioners and even poorer families may put themselves in debt to find the money so desperate are they to help their child. Tolerating this option weakens community solidarity and denies the family of ongoing support.

In democratic societies it is a political choice as how the nation’s wealth is shared but internationally, governments have acknowledged the rights of persons with disabilities to at least receive their fair share. Many have backed up this right with policies and legislation to guide the development of supports and allocate monies to fund them. Therefore, political lobbying is needed to create the policies and if they are in place, to ensure they are enacted. Globally it has been Disabled People’s Organisations and parent associations who have been the most effective lobbyists.

That said, governments often have to be prodded into acting. Local groups have fund-raised to start services with the hope that governments would take over the costs. Similarly, charitable trusts or international NGOs have instigated innovative services with an understanding that government funding would become available when the services were functioning. Additionally, governments have often delivered services especially when local politicians get involved, allied with publicity in the media and support from community leaders.

The bottom line however is that there never will be sufficient money to meet all the needs of all the people all of the time. This truism applies as much to affluent countries as to LMICs. Hence, the quest to fund services is a never-ending challenge that communities face. We take hope that working together is better than working apart in competition with one another.

## 6. Is the Seemingly Impossible Really Possible?

The cynic may dismiss much of what I have written as being impossible and at times I confess to thinking that way too. It is easy to be pessimistic and think that the child with autism will never improve, that families will break apart, that communities will never tolerate disabled people and that politicians will never care. Winston Churchill, the British leader in World War Two wrote: *A pessimist sees the difficulty in every opportunity; an optimist sees the opportunity in every difficulty*.

We have good reasons to be optimistic that the seemingly impossible can become possible. Our knowledge and understanding of how to help children with developmental difficulties is continuing to grow, most families cope admirably with the unexpected behaviours of their children, people of good will are present in every society and the world community is committed to making life better for their marginalised and disadvantaged citizens. In these pages I have described the new opportunities that can be created in response to difficulties LMICs face. However, it is not my words but our actions that make the seemingly impossible, possible. So what are we going to do?

More ‘scientific’ research per se is not the answer. Rather new forms of support have to be created in close cooperation with the community stakeholders—families in particular. Crucially the impact of the supports need to be evaluated using a range of methodologies of which participatory action research is likely to be the fore [47]. Practitioners from LMICs need greater opportunities to share their experiences with one another and ways found of overcoming the barriers they experience to acquiring new knowledge and skills [48]. Fortuitously modern technology offers opportunities for doing this which have yet to be fully exploited [49]. Finally, closer partnerships between academics and practitioners in high income countries and those based in LMICs would, I predict, result in mutual gains and reflect a truly international effort to overcome the life-limiting effects of autism [50].

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
