# Peer review of "Responding to Autism in Low and Middle Income Countries (Lmic): What to Do and What Not to Do"

_brainsci, 2022, doi:10.3390/brainsci12111475_

Round 1

Reviewer 1 Report

This is an interesting paper about autism in low and middle income countries. The paper is well written and it is of interest for the readers; however, several changes should be made before considering it for publication. 

The paper is a narrative review on several aspects of autism, and geographical implications. Despite it is a narrative review, I would recommend to change the structure of the paper.

I recommend to structure it as Introduction, Methods, Results, Discussion, Conclusions.

Abstract

1- I recommend to structure the abstract into Introduction, Aims, Methods (of the review), Results and main conclusions of the review.

Introduction

1-In the introduction section, I recommend to introduce the prevalence of autism, its frequency within the group of mental/neurological disorders, etiopathogenesis.

Results

1- The authors have mentioned "What to do and what not to do". It does implicate that they are reporting recommendations and things that are not recommended. In scientific literature, this should be accompanied by several references, mostly new, and by clinical guidelines.

I recommend to add more recommendations from clinical guidelines about the management of autism spectrum disorders in children- adolescents and adults.

What does it mean that more professionals are not the solution? Community mental health approaches are highly effective in the treatment of autism. 

Community support workers are very important. The authors have built a subsection (4.3.) about this topic. I recommend to add more references.

Turn-over of staff is really important, particularly in these patients. Is there any study investigating it in other psychiatric disorders or neurodevelopmental disorders?

The section 6 is a kind of global discussion. I think it should be expanded and reformulated. 

Reviewer 2 Report

This paper makes a great contribution to the discourse around autism interventions in low and middle income countries as an opinion piece that brings together a range of insights and perspectives. I am concerned that there is not enough citation for major statements in numerous areas within the paper to elevate the quality of the work to a publication standard. I am also concerned by the use of emotive language throughout the paper, rather than more neutral scholarly language which would provide a more nuanced and considered discourse within the paper. I appreciate the passion and commitment of the author, but suggest that a more moderate and tempered use of language would convey the ideas and messages more clearly.

Some detailed feedback follows:

The title captures the essence of the paper, however, I do not think it appropriate to include a statement suggesting that this paper identifies what works and what does not work. I would recommend only including the first section in the title as this is a narrative paper, not a data based paper.

Abstract: The abstract covers the main points identified in the article. I suggest reconsidering use of the term ASD/autism spectrum disorders and remain consistent with your title and use the term autism. This term is more accepted by the autism community and could be used throughout the paper. Page 1 line 11 you suggest that services delivered in high income countries are unsuitable for LMIC, I suggest this should be moderated to “may be” as you later indicate that there are some service types that are applicable. Line 23 the word person, should be persons…

Main Paper:

You tend to use emotive and polarising language at times, this would benefit from moderation to more clearly position your ideas and arguments in a neutral manner.

Page 2 line 66 the word where should be were. And in line 85, I suggest changing the term most to many nations are signatories….

The paragraph beginning with Equally, LMICs can bring …requires revision. You start with a focus on the contribution of LMICs and then move to the lack of service responsiveness in high income countries which is not a logical progression.

You clearly position this as a review based on experience and opinion, but this needs to be elevated to include strong links with relevant sources in order be suitable for publication. On page 3 lines119-137 you need to support these points with relevant references. This also applies to page 4, lines 182-194.

On page 5 paragraph 1, you surmise about the value of support service system for autistic children. This segment would be enhanced by an example or a reference to an existing similar service or parallel service type for another group of children. The following points 3.1-3.6 are well made and linked with relevant references. There is a typo on page 6 line 275, the word should be outcomes rather than outworking…..

The section on operational issues makes some good points and links with literature, however you have not included autistic people as part of the leadership group who have expertise in autism. This is an oversight that should be addressed.

Round 2

Reviewer 2 Report

You have addressed the major issues I identified with this paper. I accept your view that the use of emotive language is a matter of style and agree to disagree with this. I appreciate the inclusion of additional references to support the statements made on page 3 and 4. The moderation of the title softens the claims of efficacy, but remains directive. You have addressed the other issues identified.